# Single T gate in a Clifford circuit drives transition to universal entanglement spectrum statistics

Shiyu Zhou[1*], Zhi-Cheng Yang[1], Alioscia Hamma[2] and Claudio Chamon[1]

**1** Physics Department, Boston University, Boston, MA 02125, USA
**2** Physics Department, University of Massachusetts Boston, Boston, MA 02125, USA

* zhous@bu.edu

## Abstract

Clifford circuits are insufficient for universal quantum computation or creating $t$-designs with $t \geq 4$. While the entanglement entropy is not a telltale of this insufficiency, the entanglement spectrum of a time evolved random product state is: the entanglement levels are Poisson-distributed for circuits restricted to the Clifford gate-set, while the levels follow Wigner-Dyson statistics when universal gates are used. In this paper we show, using finite-size scaling analysis of different measures of level spacing statistics, that in the thermodynamic limit, inserting a *single* T ($\pi/8$) gate in the middle of a random Clifford circuit is sufficient to alter the entanglement spectrum from a Poisson to a Wigner-Dyson distribution.



# 1   Introduction

In the past few years, the dynamics of entanglement growth in non-equilibrium settings have been intensively explored, unveiling rich structures and universality classes analogous to equilibrium phenomena [1–5]. Recently, studies along this direction have been extended from entropic measures to the full entanglement spectrum (ES) [6], which captures the finer structure of entanglement. It has been shown that the dynamics of ES is able to distinguish between random unitary circuits of different complexities [7–9], as well as thermalization and localization phases of the underlying Hamiltonian [10–13]. Moreover, the onset of level repulsion in the ES signals the spreading of operator fronts, which serves as an important diagnostic of quantum chaos and information scrambling [14–16].

A crisp example that the ES reflects the complexity of the states generated by a quantum circuit is provided by the analysis of Clifford circuits. These circuits can be efficiently simulated classically and hence are not sufficient for universal quantum computation, due to restricted single-qubit rotations [17, 18]. Although Clifford circuits can generate states with the same maximal entanglement entropy as Haar random states [19], the ES of such states is either flat (for stabilizer initial states) [4,20] or Poisson distributed (for random initial product states) [8] as opposed to Wigner-Dyson (W-D) distributed as in the case of Haar random states. Moreover, as shown in [6, 8], the transition between Poisson and W-D is connected to the emergent irreversibility of random quantum circuits, which in turn is connected to the fact that the fluctuations of the maximal entanglement entropy generated by a Clifford circuit are drastically different from that of Haar random states.

Important related problems are those of derandomization, and randomized benchmarking, that is, phase retrieval, quantum state distinguishability and estimates for the rate error of quantum channels [21–28]. These tasks require the construction of a $t-$design, that is, a set of gates that reproduces the first $t$ moments of the Haar measure [29]. A random circuit of universal gates can construct a 4−design, and a random circuit based on the Clifford group can construct a 3−design but fails to be a 4−design, which is what one needs for several protocols of derandomization. It is known, though, that Clifford group generates a good approximation of a 4−design [30]. Hence, one expects that a small added perturbation – a few gates outside the Clifford set – should suffice to reach a 4-design. In particular, perturbed Clifford circuits should be able to reproduce the fluctuations of the entanglement entropy of a system evolved with a universal quantum circuit, which typically requires a higher-order design than that needed to reproduce the average entanglement entropy.

In this paper we answer the question of what density of T gates one needs to add to a Clifford circuit to alter the ES from a Poisson to a W-D distribution, a necessary condition for universal quantum circuits. Moreover, we put forward a conjecture about the transition to unlearnability and higher $t-$designs.

Consider the setup as shown in Fig. 1 (left panel). We first evolve random product states using random Clifford circuits, until their entanglement entropy reaches maximum. Then we insert a layer of T gates acting on a certain number of randomly chosen qubits into the circuit, and continue evolving with random Clifford circuits. Since the entanglement entropy has already saturated prior to the insertion of T gates, it cannot further increase. However, the ES may change following the second stage of time evolution. We ask the question: how many T gates are needed in the thermodynamic limit to alter the ES from a Poisson to a W-D distribution? Remarkably, we find, using finite-size scaling analysis of various ES statistics measures, that a *single* T gate is sufficient to poison the Poisson statistics of pure Clifford circuits in the thermodynamic limit. The deviation from W-D distribution for systems of $N$ qubits scales as $e^{-\gamma n_T N}$, where $\gamma$ is a constant of order one and $n_T$ is the number of T gates inserted. This indicates that the ES flows to W-D distribution in the infinite system size limit

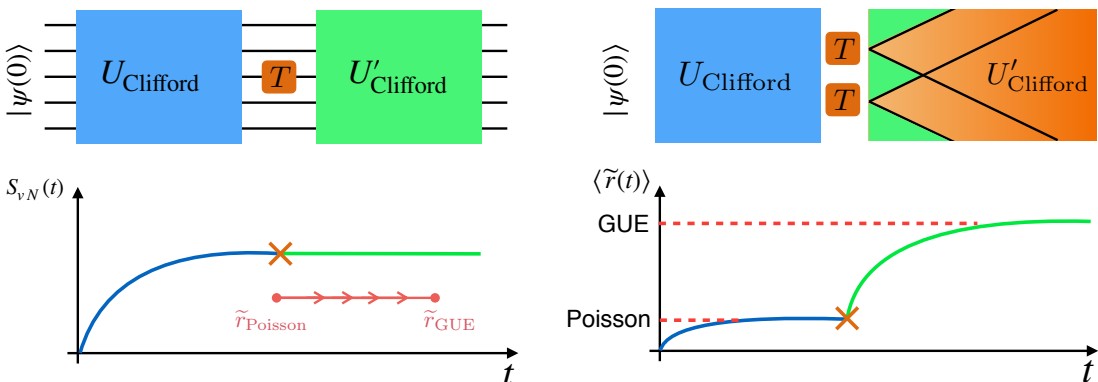

Figure 1: **Left**: A schematic of the setup considered in this work. Initial product states are first evolved under random Clifford circuits $U_{\mathrm{Cl}}$ until their entanglement entropy reaches maximum. Each circuit is constructed with 500 random non-local Clifford gates to maximize the scrambling speed. A certain number of T gates in one layer are then inserted in the circuit, followed by a second stage of random Clifford evolution $U'_{\mathrm{Cl}}$ also consisting of 500 random non-local Clifford gates. In the thermodynamic limit, the entanglement spectrum approaches a W-D distribution upon inserting a *single* T gate. **Right**: A cartoon picture showing the operator spreading of the inserted T gates during the second stage of evolution. Each T gate spreads over a spatial region that grows linearly in time (circuit depth), which sets the timescale for the saturation of the ES statistics to W-D distribution. The onset of W-D statistics occurs when the footprints of the inserted T gates cover all qubits. For this simulation, we choose instead *local* Clifford gates, so that time is measured in terms of circuit depths.

for any non-zero $n_T$. In addition, we also consider two different cases in which either: (1) the initial states are chosen as stabilizer states; or (2) the time evolution following the insertion of T gates is given by the inverse of the initial evolution. In the former case with stabilizer initial states, we find that insertion of T gates in a single layer is not sufficient to obtain W-D distributed ES, in contrast to random product initial states. The latter scenario has a natural interpretation in terms of either the noise threshold for reversibility in quantum circuits, or the operator spreading under Clifford dynamics. We find that in this case one needs order $\mathcal{O}(N^\alpha)$ ($\alpha \approx 0.6$) T gates to get W-D distributed ES (for which the density still vanishes as $1/N^{1-\alpha}$).

While we do not have an analytical proof of the above scalings, we present a physical picture that elucidates the evolution of the ES in the second stage in terms of the operator spreading of the inserted T gates, as depicted in Fig. 1 (right panel) [15, 16]. We show that each T gate spreads over a spatial region that grows linearly in time, and hence the operator spreading of T gates should determine the timescale for the saturation to Wigner-Dyson distribution of the wavefunction ES. More specifically, we find below that at a fixed system size $N$, the ES flows from Poisson towards W-D statistics as a function of the total length of the spatial region covered by the spreading of T gates: ($n_T \times \tau$), where $\tau$ is the depth of the circuit in the second stage, which supports the above picture.

## 2 Setup

We consider a "composite" quantum circuit consisting of three pieces, as depicted schematically in Fig. 1. The initial state is first evolved under a random Clifford circuit, for which the corresponding unitary operator is denoted as $U_{\mathrm{Cl}} = \prod_k U_k$, where $U_k$ is the evolution operator

at $k$-th time step in the circuit. A random Clifford circuit is constructed by picking randomly any of the following three elementary gates with equal probability at each time step: (1) H (Hadamard) gate, which takes $|0\rangle \to \frac{1}{\sqrt{2}}(|0\rangle + |1\rangle)$ and $|1\rangle \to \frac{1}{\sqrt{2}}(|0\rangle - |1\rangle)$; (2) S ($\pi/4$) gate, which gives a state-dependent phase factor: $|0\rangle \to |0\rangle$ and $|1\rangle \to e^{i\pi/2}|1\rangle$; and (3) CNOT (CONTROLLED-NOT) gate, which flips the second qubit conditioned on the state of the first one: $|00\rangle \to |00\rangle, |01\rangle \to |01\rangle, |10\rangle \to |11\rangle, |11\rangle \to |10\rangle$.

The initial state is evolved for sufficiently long time until the half-system entanglement entropy saturates to its maximal value [19]. We then randomly apply to the state $n_T \leq N$ T gates acting on $n_T$ distinct qubits. The T gate generates a single-qubit rotation about the $\sigma_z$-axis similar to the S gate, but with a different rotation angle: $|0\rangle \to |0\rangle$ and $|1\rangle \to e^{i\pi/4}|1\rangle$. We remark that replacing S gate with T gate leads to a gate set that is sufficient for universal quantum computation [18]. Finally, the state after applying T gates is further evolved with another random Clifford circuit $U'_{\mathrm{Cl}}$ with the same number of gates as $U_{\mathrm{CL}}$. We shall focus on the case where $U_{\mathrm{Cl}}$ and $U'_{\mathrm{Cl}}$ are distinct. However, we also consider in what follows a special situation where $U'_{\mathrm{Cl}} = U^{-1}_{\mathrm{Cl}}$. As we will see, the result is quite different in this special case, in contrast to a random $U'_{\mathrm{Cl}}$.

The initial states are chosen to be random product states $|\psi(0)\rangle = \otimes_{i=1}^{N}|\psi_i\rangle$, where $|\psi_i\rangle = \cos\theta_i|0\rangle + \sin\theta_i|1\rangle$ with random angles $\theta_i \in [0, \pi]$. Notice that this initial state is *not* a stabilizer state, and we will briefly comment on the situation of stabilizer initial states at the end. Under random Clifford circuit evolution, the ES $\{p_k = \lambda_k^2\}$, defined as the eigenvalues of the reduced density matrix under an equi-bipartitioning of the system $\rho_A = \mathrm{tr}_B|\psi\rangle\langle\psi|$, exhibits a Poisson level spacing distribution [8], which can be captured by the ratio of adjacent gaps in the spectrum: $r_k = (\lambda_{k-1} - \lambda_k)/(\lambda_k - \lambda_{k+1})$, with $\lambda_k \geq \lambda_{k+1}$. Poisson distributed level spacings lead to the following distribution for $r$: $P(r) = 1/(1 + r)^2$ with no level repulsion at $r = 0$. On the other hand, for levels of random matrix ensembles, the distribution of $r$ follows the W-D surmise [31]: $P(r) = (r + r^2)^{\beta}/[Z(1 + r + r^2)^{1+3\beta/2}]$, with $Z = \frac{4\pi}{81\sqrt{3}}$ and $\beta = 2$ for the Gaussian unitary ensemble (GUE). For the purpose of finite-size scaling, it is favorable to characterize the full distribution of the level-spacing ratio of the ES using a single number. Thus, we also compute a modified version of the $r$-ratio introduced above: $\widetilde{r}_k = \min\{\delta_k, \delta_{k+1}\}/\max\{\delta_k, \delta_{k+1}\}$, where $\delta_k = \lambda_{k-1} - \lambda_k$ is the gap between adjacent eigenvalues [32]. The average value $\langle\widetilde{r}\rangle \approx 0.39$ for Poisson distributed spectrum, and $\langle\widetilde{r}\rangle \approx 0.6$ for GUE distributed spectrum.

Since the entanglement entropy is already saturated to its maximum value prior to inserting T gates, it cannot further increase following subsequent time evolution. Nevertheless, the structure of the state - in particular the ES - could still evolve as the gates realizing $U'_{\mathrm{Cl}}$ are sequentially applied, due to the operator spreading of T gates. Below, we shall first construct a color map to visualize the wavefunction. This map reveals the qualitative change of structure in the bipartite entanglement following the insertion of T gates. To extract a quantitative measure, we numerically study the evolution of the ES statistics and $\langle\widetilde{r}\rangle$ as function of $n_T$ and $N$, and extrapolate to the thermodynamic limit $N \to \infty$ via a finite size scaling analysis.

## 3 Numerical results

### 3.1 Structure of the wavefunctions

We numerically simulate the time evolution protocol of the composite quantum circuit in Fig. 1 with different numbers of T gates inserted, $n_T \leq N$. Before performing a quantitative analysis, let us first visualize the structure of the amplitudes of the resultant wavefunctions in a local basis, e.g. the computational $z$-basis. We bipartition the system into subsystems $A$ and $B$, and

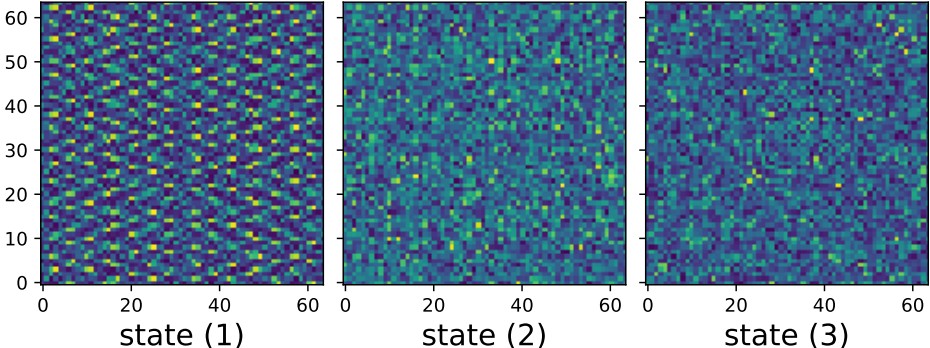

Figure 2: (Color Online) Color map of the matrix $|\Psi(x_A, x_B)|$ under a bipartitioning of the system for (1) a state evolved with a Clifford circuit; (2) a state evolved with the composite quantum circuit in Fig. 1 with $n_T = 12$ T gates inserted; (3) a random state with entries drawn from a Gaussian distribution. Blue (darker) corresponds to smaller values of $|\Psi(x_A, x_B)|$ and yellow (lighter) corresponds to larger values. The system size is $N = 12$.

the wavefunction can be written as:

$$|\psi\rangle = \sum_{x_A, x_B} \Psi(x_A, x_B)|x_A\rangle|x_B\rangle, \tag{1}$$

where $x_A$ and $x_B$ label the $z$-basis configurations of subsystems $A$ and $B$, and $\Psi(x_A, x_B)$ is the amplitude of the wavefunction recast as a matrix. We display $|\Psi(x_A, x_B)|$ by employing a color map with $x_A$ in horizontal axis and $x_B$ in vertical axis. In Fig. 2, we plot the magnitude of the amplitudes of three representative states:

(1) a state evolved with a Clifford circuit, whose ES is Poisson distributed;

(2) a state evolved under the composite quantum circuit in Fig. 1 with $n_T = N$ T gates inserted, whose ES is W-D distributed;

(3) a random state with amplitudes drawn from a Gaussian distribution, whose ES is W-D distributed.

As shown in Fig. 2, a global pattern of the wavefunction amplitudes is clearly visible for state (1), which is completely absent for a Haar random state (3). This crude measure is already capable of revealing the distinctions between states generated by Clifford circuits and Haar random states, which the entanglement entropy fails to capture. The structure in state (1) is a telltale that Clifford circuits cannot fully randomize the state, which is consistent with the absence of level repulsion in the ES. On the other hand, for state (2) obtained with a whole layer of T gates inserted, we observe no global structure like in state (1); instead it shares the structureless feature of the Haar random state (3), which indicates that Clifford circuits with few inserted T gates show properties akin to those of random unitary circuits. The striking visual difference between the states that can be prepared by action of Clifford gates and states that require universal resources suggests that such phases can be identified by machine learning architectures based on neural networks [33].

Below, we shall quantify the distinctions in the wavefunction amplitudes by studying the ES of $|\psi\rangle$, which is precisely the singular value spectrum of the matrix $\Psi(x_A, x_B)$ shown in Fig. 2. In particular, we shall address the question of how many T gates are needed in the Clifford circuit in order to achieve a W-D distributed ES, which corresponds to a randomized wavefunction.

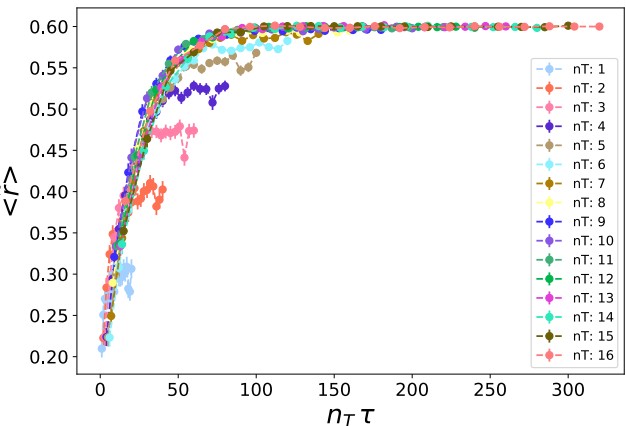

Figure 3: The average $\langle \widetilde{r} \rangle$ plotted as a function of $n_T \tau$ past the insertion of T gates, for $n_T$ ranging from 1 to 16, and $\tau$ ranging from 4 to 80 in steps of 4. The system size is fixed to $N = 16$. All curves collapse to a universal scaling function $\langle \widetilde{r} \rangle = h(n_T \times \tau)$ interpolating between $\widetilde{r}_{\mathrm{Poisson}} \approx 0.39$ and $\widetilde{r}_{\mathrm{GUE}} \approx 0.6$. The plateau for smaller $n_T$ at large $\tau$ is due to finite-size effect.

## 3.2 Finite-size scaling of the ES statistics

As we have explained previously, we choose the particular setup shown in Fig. 1 such that the entanglement entropy already saturates before inserting the T gates. Therefore, the addition of T gates has no effect on the *amount* of entanglement generated by the quantum circuit. Nevertheless, the ES spectrum can still change, and in particular, level repulsion can emerge due to the spreading of the downstream effects of inserted T gates in the second stage of Clifford evolution $U'_{\mathrm{Cl}}$ as illustrated in Fig. 1. We shall first study how the ES transitions from Poisson to W-D distribution by looking at the time dependence of $\langle \widetilde{r}(\tau) \rangle$ past the insertion of T gates. Time $\tau$ is measured in terms of the circuit depth.

In Fig. 3, we plot the average $\langle \widetilde{r}(\tau) \rangle$ for a fixed system size $N = 16$ and varying $n_T$. (In this study, we use a 1-dimensional circuit brick wall layout.) We find that the curves for different $n_T$ collapse to a universal scaling function:

$$\langle \widetilde{r}(\tau) \rangle = h(n_T \times \tau). \tag{2}$$

This scaling form can be understood as follows. The downstream effect the action of the inserted T gates is contained within a light-cone, such that each T gate covers a spatial region of size $\xi \sim \tau$ at time $\tau$. If $n_T$ T gates are inserted, the scale of the footprint of the region affected by the T gates is $n_T \xi \sim n_T \tau$. Hence, the crossover to GUE statistics should occur when the footprint of the affected region covers all qubits, i.e., when $n_T \tau \sim N$. This relation indicates that the larger $n_T$, the shorter time it takes to reach the asymptotic (GUE) value of $\langle \widetilde{r} \rangle$. Note that in Fig. 3 at small $n_T$, $\langle \widetilde{r} \rangle$ saturates to a plateau value less than the GUE value and peals off from the universal scaling function. This deviation is due to finite size effects. In finite system sizes, time evolution of only a small amount of T gates cannot fully randomize the states, and hence the infinite-time – in practice, $\tau \gg N$ – average level spacing ratio $\langle \widetilde{r}(\tau \to \infty) \rangle$ remains below that of a W-D distribution. To see what happens in the thermodynamic limit where the system size $N \to \infty$, we shall now focus on the finite-size effect by looking at the infinite-time average level spacing ratio as a function of $n_T$ and $N$: $\langle \widetilde{r}(\tau \to \infty, n_T, N) \rangle$.

In Fig. 4 (left panel), we plot the infinite-time $\langle \widetilde{r} \rangle$ for various system sizes and numbers of T gates inserted. Remarkably, we find that all curves again collapse to a universal scaling

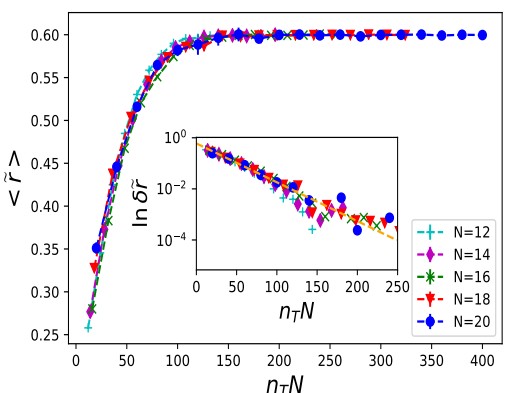
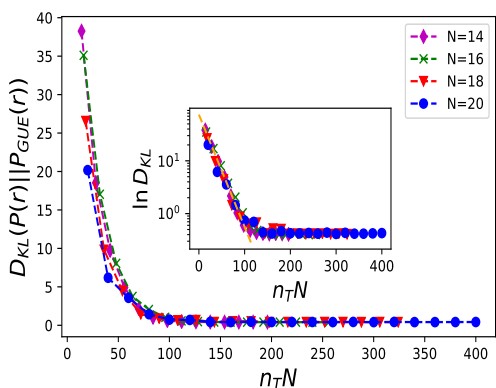

Figure 4: Finite-size scaling of the infinite-time ES. **Left**: The infinite-time average level spacing ratio $\langle \widetilde{r}(\tau \to \infty) \rangle$ versus $n_T N$, for system sizes $N = 10, 12, 14, 16, 18$ and 20. All curves collapse to a universal scaling function $\langle \widetilde{r}(\tau \to \infty) \rangle = f(n_T N)$ interpolating between $\widetilde{r}_{\text{Poisson}} \approx 0.39$ and $\widetilde{r}_{\text{GUE}} \approx 0.6$. Inset: deviations from $\widetilde{r}_{\text{GUE}}$ in log-linear scale, indicating the exponential scaling form $\delta \widetilde{r} \approx \widetilde{r}_0 e^{-\gamma n_T N}$. **Right**: The KL divergence between the full ES level spacing distribution and the W-D distribution. $D_{\text{KL}}$ for different curves also collapses to a universal scaling function of the product $n_T N$. Inset: $D_{\text{KL}}$ decays exponentially with $n_T N$, similarly to $\delta \widetilde{r}$. The data are averaged over 3000 ($N = 10$), 2000 ($N = 12$), 1500 ($N = 14$), 1000 ($N = 16$), 500 ($N = 18$), and 150 ($N = 20$) realizations. When not visible, the error-bars are smaller than the size of the data points.

function of the form:

$$\langle \widetilde{r}(\tau \to \infty) \rangle = f(n_T \times N). \tag{3}$$

In other words, $\widetilde{r}$ is only a function of the product $n_T N$. Moreover, we find that the deviation from $\widetilde{r}_{\text{GUE}}$ takes the following form:

$$\delta \widetilde{r} \approx \widetilde{r}_0 \, e^{-\gamma \, n_T N}, \tag{4}$$

with some constants $\widetilde{r}_0$ and $\gamma$, as shown in the inset of Fig. 4 (left panel). One immediately concludes that in the thermodynamic limit, as long as $n_T \neq 0$, $\langle \widetilde{r} \rangle = f(\infty) = \widetilde{r}_{\text{GUE}}$, that is, all that one needs in the thermodynamic limit is a *single* T gate to change the ES to random matrix theory behaviors! A single T gate inserted into the Clifford circuit acts like a "poison pill" that completely randomizes the final state and kills the Poisson distribution. This can be understood as follows. Although the density of T gates is vanishing, it nevertheless changes the *global* phase structure of the quantum state. The ES is a global property of the full wave function, hence the effect of even a single T gate is not negligible. Note that the $n_T N$ scaling is consistent with the plateau values in Fig. 3, which corresponds to $\langle \widetilde{r}(\tau \to \infty) \rangle$ at fixed $n_T$ and $N$.

Although the average value of $\langle \widetilde{r} \rangle$ reaches that of GUE in Fig. 4 (left panel), it only characterizes the first moment of the full distribution. To further substantiate that the ES level spacing statistics follows a W-D distribution, we numerically compute the distance between the full distribution of the ES level-spacing ratio $r$ of the final states and the W-D distribution (GUE in particular), defined as the Kullback-Leibler (KL) divergence:

$$D_{\text{KL}}[P(r) \| P_{\text{GUE}}(r)] = \sum_i P(r_i) \ln [P(r_i)/P_{\text{GUE}}(r_i)]. \tag{5}$$

As shown in Fig. 4 (right panel), the KL divergence between the ES level spacing distribution and W-D distribution also collapses to a universal scaling function of the product $n_T N$. Therefore, the KL divergence also goes to zero in the thermodynamic limit for any nonzero $n_T$, confirming that even the full ES level spacing distribution reaches the W-D distribution with the insertion of a single T gate. The full distribution $P(r)$ corresponding to a particular point where $\langle \tilde{r} \rangle = \tilde{r}_{\text{GUE}}$ is presented in Fig. 5 (left panel), showing that the full distribution of ES level spacing statistics indeed follows W-D.

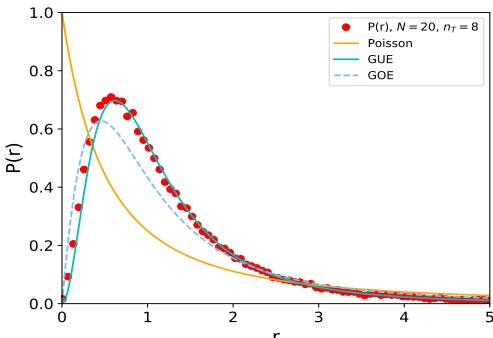

Figure 5: The ES level spacing distribution $P(r)$ for $N = 20$ with $n_T = 8$ T gates inserted.

## 3.3 A composite quantum circuit with $U'_{\text{Cl}} = U_{\text{Cl}}^{-1}$

So far we have been considering the general situation where the Clifford circuits before and after the insertion of T gates (i.e., $U_{\text{Cl}}$ and $U'_{\text{Cl}}$ in Fig. 1) are different and uncorrelated. It is nonetheless interesting to look at a special case in which $U'_{\text{Cl}} = U_{\text{Cl}}^{-1}$, i.e the Clifford evolution in the second stage is the exact inverse of the initial evolution operator. This particular example is interesting for several reasons. In the absence of any T gate inserted in the middle, $U_{\text{Cl}}$ and $U_{\text{Cl}}^{-1}$ will cancel exactly, bringing the final state back to the original product state. However, this cancellation ceases to happen as more and more T gates are inserted in between. If one views the insertion of T gates as a particular kind of noise in the circuit, this example resembles the noise threshold for reversibility in a quantum circuit. Second, the composite quantum circuit in this particular case coincides with the spreading of T operators under random Clifford dynamics in the Heisenberg picture. How many T gates are needed such that their spreading under random Clifford circuit evolution is sufficient to generate W-D distributed ES when acting on a random product state?

In Fig. 6, we plot the infinite-time average $\langle \tilde{r}(\tau \to \infty) \rangle$ for different system sizes and numbers of T gates inserted. In contrast to the previous case, here we find instead that different curves collapse to another universal scaling function:

$$\langle \tilde{r} \rangle = g(n_T / N^\alpha), \tag{6}$$

where $\alpha > 0$, and we numerically find $\alpha \approx 0.6$. We emphasize that the precise value of $\alpha$ is not the main focus of our study. Rather, the fact that $\alpha$ must be positive is the most important message here, which is in sharp contrast to Eq. (3). The difference between Eq. (6) and Eq. (3) drastically changes the behavior in the thermodynamic limit. From Fig. 6, we find that the average $\langle \tilde{r} \rangle$ reaches $\tilde{r}_{\text{GUE}}$ when $n_T / N^{0.6} \sim \mathcal{O}(1)$. Therefore, in the infinite system size limit, one needs number $n_T \sim \mathcal{O}(N^{0.6})$ T gates to alter the ES from Poisson to W-D distribution. Notice that even in this case, since $n_T$ is only sub-extensive in system size, the required density of T gates still vanishes as $1/N^{0.4}$.

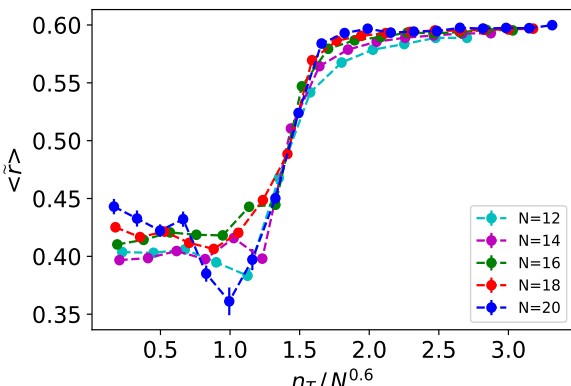

Figure 6: The infinite-time average $\langle \widetilde{r}(\tau \to \infty) \rangle$ versus $n_T/N^{0.6}$ for the special case in which $U'_{\mathrm{Cl}} = U^{-1}_{\mathrm{Cl}}$, for systems sizes $N = 12, 14, 16, 18$ and 20. All curves collapse to a universal scaling function $\langle \widetilde{r} \rangle = g(n_T/N^{0.6})$ interpolating between $\widetilde{r}_{\mathrm{Poisson}} \approx 0.39$ and $\widetilde{r}_{\mathrm{GUE}} \approx 0.6$. The numbers of realizations are the same as in Fig. 4. When not visible, the error-bars are smaller than the size of the data points.

To elucidate how the insertions of the T gates alter the ES in the present case, let us consider explicitly their spreading:

$$
\begin{aligned}
U_{\mathrm{Cl}} \left( \prod_{k=1}^{n_T} T_{i_k} \right) U^{-1}_{\mathrm{Cl}} &= U_{\mathrm{Cl}} \left[ \prod_{k=1}^{n_T} \left( \cos\theta \, \mathbb{1} + i\sin\theta \, Z_{i_k} \right) \right] U^{-1}_{\mathrm{Cl}} \\
&= \prod_{k=1}^{n_T} \left( \cos\theta \, \mathbb{1} + i\sin\theta \, \widetilde{Z}_{i_k} \right),
\end{aligned}
\tag{7}
$$

where $\theta = \frac{\pi}{8}$ for the T gate, and $Z_{i_k}$ is the Pauli-$Z$ operator acting on site $i_k$. In the second line of Eq. (7), we have used the property that, under Clifford dynamics, a Pauli operator evolves into a single string of Pauli operators $\widetilde{Z}_{i_k}$ rather than a superposition of Pauli strings [18]. Upon averaging over circuit realizations, one expects that the $\widetilde{Z}_{i_k}$ are essentially random Pauli strings up to the fact that they remain commuting with one another, regardless of the original positions of the inserted T gates. When acting on a product state, each Pauli string operator $\widetilde{Z}_{i_k}$ (and the products of which) simply produces another product state, with each qubit being flipped or not depending on the particular type of Pauli operator on each site. Therefore, the time-evolved operator in the above equation will generate a superposition of $2^{n_T}$ product states, when applied to a random product state, and the entanglement entropy is thus upper-bounded by

$$
S_{vN} \leq n_T \ln 2.
\tag{8}
$$

In this situation the insertion of $\mathcal{O}(1)$ T gates is insufficient to discern statistical properties of the ES, since the rank of the density matrix is too low to yield a spectrum with enough non-vanishing eigenvalues. This is also the reason why the data shown in Fig. 6 are noisier at small $n_T$. Therefore, when $U'_{\mathrm{Cl}} = U^{-1}_{\mathrm{Cl}}$, one needs a subextensive number of T gates to alter the ES.

In Fig. 7, we plot the ES statistics of states generated by applying the operator in Eq. (7) to random product states, but with the specific $\widetilde{Z}_{i_k}$ operator associated with the spreading of $Z_{i_k}$ replaced by a *random* string of Pauli operators. On the left panel, we use $\theta = \frac{\pi}{8}$, corresponding to insertion of T gates, and find that the ES is W-D distributed, in agreement with what was obtained by directly simulating the composite random circuit evolutions. In contrast, if we use $\theta = \frac{\pi}{4}$ instead, corresponding to insertion of S gates, the ES exhibits a Poisson distribution shown in the right panel, which is consistent with known results for random Clifford

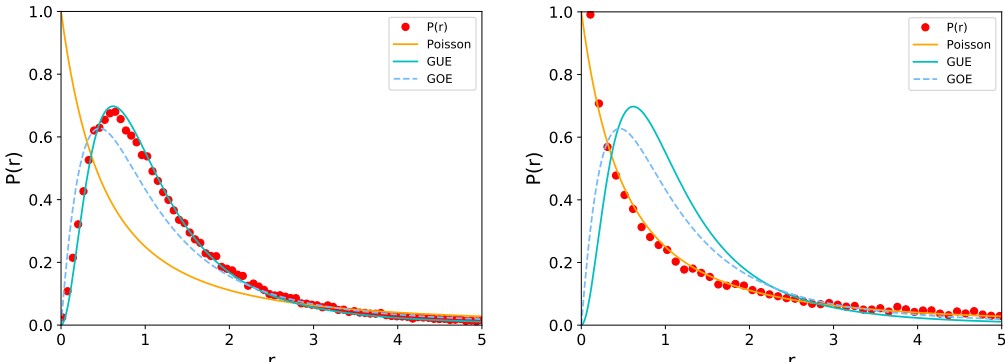

Figure 7: ES statistics of states generated by applying Eq. 7 to random product states. Left: W-D distribution for $\theta = \frac{\pi}{8}$ (inserting T gates); right: Poisson distribution for $\theta = \frac{\pi}{4}$ (inserting S gates). The data are obtained for system size $N = 12$ with $n_T = 12$ T or S gate inserted, and averaged over 800 realizations of initial states and random Pauli strings.

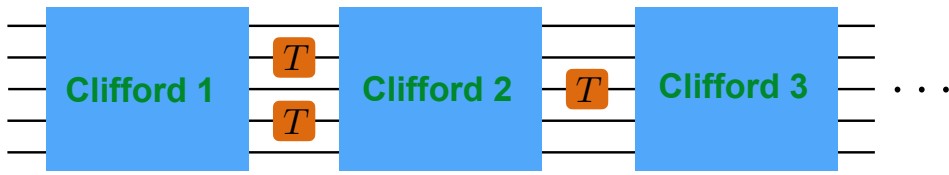

Figure 8: A potential architecture for universal quantum computing, where only a small number of T gates are added at largely spaced layers.

circuits [8]. We thus conclude that replacing the specific $\widetilde{Z}_{i_k}$ operator associated with $Z_{i_k}$ by a *random* string of Pauli operators has no effect in the ES. Instead, it is the *angle $\theta$* associated to the specific single-qubit rotation that alters the relative amplitudes of the superposition of $2^{n_T}$ product states to yield the ES associated with the gate set.

We remark that the above results hold when we choose as initial states random product states, where each qubit points along different directions on the Bloch sphere to begin with. Instead, if one starts from stabilizer states, for which random Clifford dynamics yields a flat ES as opposed to Poisson, insertion of T gates in the same manner as in Fig. 1 does *not* lead to W-D distributed ES. Therefore, the random angles imprinted in the initial states, although by themselves cannot produce Haar random states under Clifford circuit evolution, turns out to be essential for the T gates to wipe off the Poisson distributed ES.

## 4 Conclusions

In this work, we study the number of T gates needed in a random Clifford circuit to alter the ES statistics from Poisson to W-D distribution, a necessary condition for the underlying circuit to be universal. We construct a composite quantum circuit as in Fig. 1 with T gates inserted in the middle, and show that a single T gate is in fact sufficient to obtain W-D distributed ES, as is the case for Haar random states. Our results suggest that the density of T gates needed for universal quantum computation might be vanishing in the thermodynamic limit. Given the difficulties in realizing T gates within various experimental platforms for quantum

computing (e.g. topological quantum computing based on Majorana zero modes), we propose an alternative construction of quantum circuits where one may be able to generate a good enough approximation to a random unitary by concatenating segments of Clifford evolutions with very few T gates inserted at largely spaced layers, as illustrated in Fig. 8. While this may lead to an overhead in the circuit depth, it is advantageous in cases where implementations of T gates are hard.

In closing, we would like to ask the question of whether the transition driven by the density of non-Clifford gates also characterizes the transition to 4−designs and unlearnability of random quantum circuits. As shown in [7,8], the transition to W-D for ES level spacing statistics corresponds to the impossibility of reversing the circuit by means of a Metropolis-based disentangling algorithm. For Poisson-distributed ES, reversal is possible, and this corresponds to the possibility of learning, via disentangling, a circuit that takes the given product state to the entangled output state. Therefore, the onset of irreversibility is also a learnability-unlearnability transition. We conjecture that learnability is due to the structure of temporal fluctuations in 2−Rényi entropy. This would imply that the transition to the W-D ES is also a transition from 3−design to (at least) 4−design. The Clifford group falls short of being a 4−design but it is a very good approximation of it [30]. In particular, it has the same 2−Rényi entropy [34] but not the same fluctuations. We conjecture that the transition to 4−design, W-D and unlearnability are one and the same.

*Note Added*: The conjecture we put forward in this paper about the number of T gates needed to drive a Clifford circuit to a 4-design has been recently proved in arxiv:2002.09524.

# Acknowledgements

We thank Thomas Iadecola for useful comments on the manuscript. This work was supported by the U.S. Department of Energy (DOE), Division of Condensed Matter Physics and Materials Science, under Contract No. de-sc0019275.

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
