# Peer review of "Single T gate in a Clifford circuit drives transition to universal entanglement spectrum statistics"

_SciPost Physics, doi:SciPost Phys. 9, 087 (2020)_

## Round 1 · Referee Report · Anonymous · 2020-5-9

Report

Clifford circuits are a restricted class of quantum circuits. They are not sufficient for universal computation (their action on a restricted class of initial states can be simulated classically) but nevertheless they are important in quantum information, for example as starting points for the addition of extra gates to give universality. This paper studies an aspect of this crossover from Clifford circuits to more generic behavior on addition of a small number of non-Clifford gates.

This is an interesting question, and the paper provides some interesting results from computer simulations, showing that one quantity (r value of entanglement levels) returns to generic behavior even with only a single non-Clifford gate. They also use this to motivate a conjecture for more general quantities, albeit without much detail.

On the other hand, the paper is limited to this numerical obervation, with limited extensions. An analysis which might give understanding of the observation is lacking. The manuscript is also vague or imprecise in several places as described below. Therefore, to the extent that I can gauge the level required for a SciPost acceptance this paper in its present form is not substantial enough to meet this level.

Comments:

Discussion regarding Figure 2: This is very qualitative. For example we find the extremely vague statement “adding T gates into Clifford intrinsically alters Clifford circuit’s computational power, and drives the system toward randomness”.

Similarly in the introduction “The Clifford group generates a good approximation of a 4-design” is explained as “In other words, it takes very little for the Clifford group to become something that is capable of reproducing the fluctuations of observables evolved with a universal quantum circuit.” This is very vague. The second sentence does not explain the meaning of the first.

Equation 1: It is claimed that this is a universal scaling form. The scaling variable is claimed to be n N where n is the number of T gates and N is the number of qubits. This does not sound quite correct. In order to have a finite (”order 1”) value of the scaling variable we see that N must also be finite and “order 1”. But it is unlikely a universal scaling form can apply for a finite, order 1 number of qubits. More likely, universality only holds in the limit of asymptotically large scaling variable. This is not what is usually meant by a “scaling form”. This also applies to the comment about DKL on p5.

Bottom of p4: The authors talk of a “poisson fixed point”. It is not clear in what sense this is meant. Is there an RG scheme in mind?

p6: “this example gives the noise threshold for reversibility”. It is not clear from the text whether this a conjecture or a confirmed result.

Eq 3: The objection above does not apply to this scaling form, because it is possible to take N to infinity while fixing the value of the scaling variable. However, the evidence for this scaling form from Fig 5 is not obviously 100% conclusive. For example the absence of a theoretical argument one could imagine that the right power of N is not exactly 0.5.

p7: Here we find an analytical result for the case where the two Clifford unitaries are inverse. It is nice to have a partial analytical understanding but it would be desirable to understand how many T gates are needed. Can the authors derive the sqrt N that they assert?

p8: “this result points to a new direction…” The authors suggest that their result implies a compressibility property of circuits. Why should this be true? Can the authors provide an argument, or is it a speculation?

Final paragraph of paper: A conjecture is stated but the motivation for this conjecture is compressed to a very few lines. Can the logic be made clearer?

Minor points:

Abstract: “the entanglement spectrum” could be more specific, e.g. “of a time-evolved random product state”

Introduction p1: “derandomization” is mentioned. Could it be explained what this means?

p2. In the initial definition of the protocol “random product states” are mentioned but further definition of these states is required. For example one type of random product state would also be a stabilizer state and this is not what the authors are talking about in this definition.

I cannot find a specification of the depth of the circuits used for the data.

Bottom of p3. Define xA and xB.

Middle of p5. “the full distribution of the ES”. The distribution of the quantity r is meant, not of the full spectrum.

Below eq 4. “essentially random Pauli strings”. During the evolution these strings retain properties of commutativity and independence.

p7 “these results indicate”. The content of this sentence and the next sentence are repeated in the following two.

  • validity: -
  • significance: -
  • originality: -
  • clarity: -
  • formatting: -
  • grammar: -

Author:  Shiyu Zhou  on 2020-11-03

(in reply to Report 1 on 2020-05-09)
Category:
remark
answer to question
correction
pointer to related literature

**Figures are in the attached pdf file. **

(1.1). Indeed the discussion of Fig. 2 (in paper) was meant to be qualitative. Its point is to provide a visualization of the structures of different wavefunctions that all have maximal amount of entanglement before quantifying their distinctions using the entanglement spectrum statistics. The most intuitive way of seeing that the time-evolved states under Clifford circuits are not random states is by looking directly at their amplitudes in a given basis, which is what we show in Fig. 2 (in paper). We find from Fig. 2 (in paper) that a state evolved with a Clifford circuit with one layer of T gates inserted exhibits similar structure to a random state whose amplitudes are drawn from Gaussian distribution.

We edited the sentence that the referee mentioned, which now reads: "which indicates that Clifford circuits with inserted T gates within show properties akin to those of random unitary circuits."

(1.2). We edited the second of the two sentences that the referee listed. The second sentence was not meant to explain the first; the precise explanation of the statement of the first sentence is contained in the reference at its end. Our intention was to state with the second sentence that, since the Clifford group already generates a good approximation of a 4-design in the sense of Ref. 22 (in paper), then it should only take a small added perturbation (a few gates outside the Clifford set) to reach a 4-design. The revised sentence, we hope, is an improvement over the previous one, and makes the point more clear.

(1.3). ** Fig. 1 is in the file attached**
The scaling form in Eq. 3 (in paper) is extracted directly from the data. To clarify the origin of the scaling form, we include below, in Fig. 1 of this reply, the unscaled data for $\langle \widetilde{r} \rangle$ as function of $n_T$ for different values of $N$. In Fig. 1 (left panel) one observes that the curves are consistently shifted horizontally to the left when $N$ is increased. That the curves are shifted to the left with increasing $N$ suggests a scaling form with a variable $n_T/ N^\alpha$, with $\alpha<0$. (Had the curves shifted to the right with increasing $N$, the scaling would instead need $\alpha>0$.) The data shown in the paper collapses the curves for different $N$ with the choice $\alpha=-1$, i.e., with $n_T N$.

In Fig. 1 (right panel) we show, for a given number $n_T$ of T gates, that when $N$ increases the average $\langle \widetilde{r} \rangle$ increases with $1/N$. The finite size scaling with a linear fit shows that the $r$ value corresponding to the GUE distribution is reached in the thermodynamic limit when $n_T$ is fixed, consistent
with our scaling of $\langle \widetilde{r} \rangle$ with the variable $n_T N$.

We hope that we clarified to the referee the scaling analysis leading to $ \langle \widetilde{r} \rangle = f(n_T \, N)$.

(1.4). The words flow and fixed points pertain to the differential equation that Eq. 4 (in paper) satisfies, which appeared in the text. The associated flow contains two fixed points, one for $n_T=0$, suggesting that only in the absence of T gates one avoids the GUE statistics associated with the flow to $\delta\widetilde{r}\to 0$ when $n_T\ne 0$. While it is suggestive that this flow equation may have an RG justification, we avoided reference to RG flow specifically because we cannot substantiate it at this point.

To avoid any confusion, we removed the differential equation associated to Eq. 4 (in paper), and the discussion of its flows.

(1.5). It is a result that stands on top of a conjecture: 1) the result is the value of the threshold number of T gates to reach a Wigner-Dyson-distributed entanglement spectrum; 2) the conjecture (of Ref. 7) is that a Wigner-Dyson distributed entanglement spectrum is the signature of chaos and irreversibility. This is a conjecture, much as the connection between quantum chaos and Wigner-Dyson distributed energy spectra is (still, after three decades) a conjecture.

(1.6). ** Fig. 2 is in the file attached**
We refer to our reply to point 1.3 above to counter the referee's mentioned objection to the scaling form for the case $U'_{\rm Cl} \neq U^{-1}_{\rm Cl}$.

For the case $U'_{\rm Cl} = U^{-1}_{\rm Cl}$, the quality of the data collapse is not sufficient for us to extract the exact exponent. An exponent of $\alpha = 0.6$ is consistent with the numerical data, and we realized upon addressing the referee's question that this value provides a better fit than the original 0.5. Indeed, we do not have an analytic argument for an exponent of 0.5 or 0.6; but the numerical data is clear that there is a good collapse. We show in Fig. 2 (a) in this reply the unscaled data for the different sizes. Notice that, as opposed to the data in Fig. 1, the curves shift to the right with increasing $N$, signaling that a scaling $n_T/N^\alpha$ requires $\alpha >0$. In Fig.2 (b) and (c) we show scalings with $\alpha =0.5$ and $\alpha =0.7$, respectively. We notice that the scaling now shown in the paper with $\alpha=0.6$ is slightly better than both. In any case, we can conclude that the exponent $\alpha$ is positive.

Our data highlights the striking difference between the two cases $U'_{\rm Cl} = U^{-1}_{\rm Cl}$ and $U'_{\rm Cl} \neq U_{\rm Cl}$, where in the former a sub-extensive number of T gates is needed to push
the system to the universal GUE statistics, while in the latter a single T gate suffices as we uncover from the scaling. This finding, not the precise scaling exponent for the collapse, is the main result of our work that deserves attention.

(1.7). We agree with referee that it would be desirable to derive an analytical result for the number of T gates needed to reach GUE statistics. While it is possible to gain some intuition from considering the spreading of the Pauli strings, and the effect of the rotation angle $\theta$ in Eq.(7) (in paper) as it is varied away from $\theta=\frac{\pi}{4}$, we do not have a clear handle in the analytical calculation of the entanglement spectrum statistics, which is even harder than that of the entanglement entropy. So, unfortunately, we cannot address this criticism.

(1.8). Perhaps the word ``compress'' was a rather unfortunate choice; we are not compressing the circuit, but instead proposing to use of the T gates in small, well spaced doses. What we really mean here is that one may be able to achieve generic unitaries using fewer T gates than one would naively expect. This is particularly important in
quantum computing platforms (e.g. Majorana-based topological quantum computation) where the implementation of T gates is either hard or not protected. To be clear, this claim is speculative.

In the revised manuscript, we have changed the sentence to ``we propose an alternative construction of quantum circuits where one may be able to generate a good enough approximation to a desired random unitary by concatenating segments of Clifford evolutions with very few T gates inserted at largely spaced layers.''

(1.9). As we have shown previously in Refs.[1,2], states with a Wigner-Dyson distributed entanglement spectrum cannot be efficiently disentangled using a Metropolis algorithm, whereas states with a Poisson distributed entanglement spectrum can, even when these two classes of states both have maximal von-Neumann entropy as well as maximal Renyi-2 entropy. Alternatively, one can view the disentangling process as ``learning" a quantum circuit capable of taking the given product state to the entangled output state. In the revised manuscript, we explain this connection of learnability of a circuit to the capability of disentangling. Explicitly, we now write ``As shown in [1,2], the transition to W-D for ES level spacing statistics corresponds to the impossibility of reversing the circuit by means of a Metropolis-based disentangling algorithm. For Poisson-distributed ES, reversal is possible, and this corresponds to the possibility of learning, via disentangling, a circuit that takes the given product state to the entangled output state. Therefore, the onset of irreversibility is also a learnability-unlearnability transition.''

The conjecture that we make in the last two sentences of the paper is informed by this connection between learnability and reversibility explained above. From the result of Ref. [3], Clifford group, being a unitary 3-design, is indeed capable of producing maximal Renyi-2 entropy. It is therefore consistent with our results that the Renyi-2 entropy cannot differentiate between learnability and unlearnability of quantum circuits, and that one needs to look at the ES. However, based on our previous work, although the average Renyi-2 entropy of Clifford circuit is identical to that of random unitary circuits, its temporal fluctuations are not. Therefore we conjecture that the temporal fluctuations of Renyi-2 entropy is able to tell that the circuit is in fact only a unitary 3-design. Since the temporal fluctuations of Renyi-2 entropy is also tied to the efficiency of the disentanling algorithm and thus to the learnability transition, we further conjecture that the transitions from (i) Poisson to Wigner-Dyson distributed ES; (2) learnabibility to unlearnability; (3) unitary 3-design to (at least) unitary 4-design, are the same.

Minor points:

(1). We thank the referee for the suggestion. We have changed this in the revised manuscript.

(2). In the revised manuscript, we have added a reference to derandomization that discusses the concept.

(3). The definition of the random product states is introduced in the third paragraph of the Setup section.

(4). In the revised manuscript, we have added the information on the circuit depth in the caption of Fig. 1. In all data presented in the paper, we always choose a depth sufficient to reach maximal entanglement.

We remark that we added to the paper a study of $\langle \widetilde{r} \rangle$ as function of the depth past the insertion point of the T gates in Sec. 3.1, and illustrated in the new Figs. 1 and 3 in the revised manuscript. [Please see reply (2.2) to Referee 2 below.]

(5). $x_A$ and $x_B$ label the $z$-basis configuration of subsystem $A$ and $B$, which are used as the basis for the wavefunction components.

(6). We thank the referee for pointing this out. We have changed it to ``the full distribution of the ES level-spacing ratio $r$''.

(7). That is correct, indeed the string commutation relations and linear independences are preserved under time-evolution. These restrictions do not alter the results in our studies of random strings, but they should be noted in the text. We thank the referee for this comment.

(8). We have corrected the repetition in the revised manuscript.

[1] C. Chamon, A. Hamma, and E. R. Mucciolo, “Emergent irreversibility and entanglement spectrum statistics,” Phys. Rev. Lett. 112, 240501 (2014).
[2] D. Shaffer, C. Chamon, A. Hamma, and E. R. Mucciolo, “Irreversibility and entanglement spectrum statistics in quantum circuits,” J. Stat. Mech.: Theory and Experiment 2014, P12007 (2014).
[3] Z.-W. Liu, S. Lloyd, E. Y. Zhu, and H. Zhu, “Generalized entanglement entropies of quantum designs,” Phys. Rev. Lett. 120, 130502 (2018).

Attachment:

figures.pdf

---

## Round 1 · Referee Report · Anonymous · 2020-5-29

Report

The paper considers the question of how a non-universal, so-called Clifford circuit evolution crosses over to more generic dynamical behavior when a small number of additional non-Clifford gates are added to the circuit. In particular, the authors consider the behavior of the level statistics of the entanglement spectrum (ES), which has been shown previously to distinguish between Clifford dynamics (Poisson statistics) and chaotic evolution (Wigner-Dyson). They consider adding a single layer of local rotations that lie outside the Clifford group, preceded and followed by many layers of Clifford gates. They show numerical evidence that in the thermodynamic limit, even a single such rotations is sufficient to drive the transition to Wigner-Dyson statistics, showcasing an extreme sensitivity of the Clifford behavior.

I have found the basic question of the paper to be quite interesting and I believe it is of importance for both the quantum dynamics and the quantum computation communities. Moreover, I found the manuscript to be quite well written and easy to read, although slightly vague at times (in some cases it is unclear if certain statements are meant only to capture some intuition or something more rigorous). On the other hand, the actual results contained in the paper are somewhat limited. The authors consider only one particularly sensitive aspect of the dynamics, and even there they provide a relatively small amount of details from numerical calculations, while an analytical understanding, or a more thorough numerical exploration is missing. As such, I believe that the authors should elaborate on various points before the paper can be published.

In particular, there were quite a few questions that came to mind while reading the manuscript that are not addressed:

- The most striking statement, that a single T gate is enough to drive the transition, is somewhat insufficiently supported. The authors propose a scaling form as a function of the number of T gates (n_T) and the total system size (N), but it is unclear whether such a scaling form is meaningful in the limit when n_T = 1. I would find it very useful to have some data on how the level statistics changes when n_T is fixed and N is increased. Also, it would be good to see the un-scaled data from Figs. 3 and 5 for comparison.
- How long does it take for the ES statistics to saturate to its Wigner-Dyson form after the T gates are applied? What can be said about the way this limit is approached?
- How do the results depend on the time when the T gates are applied? What if they occur before the total entanglement saturates (as would be relevant in the thermodynamic limit)?
- Fig. 2 presents a color plot of the time-evolved wavefunction in various cases. While this is visually appealing, its physical meaning is quite unclear. Is there any meaningful quantitative statement one could make about this object? Has it been studied before? How much does the apparent structure in the Clifford case depend on the choice of basis?
- It would be very useful to have at least some qualitative analytical understanding of why there should be a scaling collapse of the form n_T * N.

A few other minor points:
- On p. 3 the definitions of S and T gates are the same. I believe the former is a typo.
- It would be good to state the number of Clifford layers applied in the numerical simulations.
- There is a repeated sentence in the next to last paragraph before the conclusions: "replacing the specific Z_ik..."

  • validity: -
  • significance: -
  • originality: -
  • clarity: -
  • formatting: -
  • grammar: -

Author:  Shiyu Zhou  on 2020-11-03

(in reply to Report 2 on 2020-05-29)
Category:
remark
answer to question
correction
pointer to related literature

** Figures are attached in the pdf file**

(2.1). ** Fig. 1 is in the attached file.**
This question largely overlaps with that raised by Referee 1, so we also refer the referee to our reply (1.3). Following the request of referee 2, we show the unscaled data in Fig. 1 of this reply.

In particular, in the right panel of Fig. 1 we show a finite size scaling to extract the asymptotic value of $\langle \widetilde{r} \rangle$ as a function of $1/N$. The right panel of Fig. 1 shows that the value of $\langle \widetilde{r} \rangle$, for $n_T=4$, increases with decreasing $1/N$, and the linear extrapolation is consistent with the GUE value being reached in the asymptotic limit.

(2.2). We thank the referee for the question of how long it takes to reach the GUE limit for $\langle \widetilde{r} \rangle$. In answering it we have developed a better understanding as to the mechanism for the crossover from one statistics to the other, and why having more T gates accelerates the process.

It has been shown in Ref. 4 that the onset of level repulsion in the ES is related to the spreading of operator front. We thus believe that operator spreading of T gates essentially sets the timescale for the saturation to Wigner-Dyson distribution in our setup. To test this hypothesis, We added to the paper a study of the time dependence of $\langle \widetilde{r} \rangle$ past the insertion of the T gates, in a 1D bit array. (Time $\tau$ is measured in terms of layers of gates acting on bits on a line.) The results of these studies are presented in Fig. 3 (in paper) of the revised paper, for $N=16$. The plot confirms that $\langle \widetilde{r}\rangle$ indeed scale as a function of $n_T\tau$. (The plateau shown for large $\tau$ for different system sizes are due to finite-size effect, and the plateau values are consistent with the $n_T N$ scaling in the previous studies.)

In Fig. 1 (right panel) of the paper we added a cartoon picture of how the ES switches from Poisson to Wigner-Dyson distributed with the spreading of the downstream effects of the inserted T gates. For the 1D circuit that we study in Fig. 3 (in paper), the spreading is linear in $\tau$, i.e., each T gates covers a region of size $\xi \sim \tau$ along the bit line. If $n_T$ gates are inserted, the scale of the footprint of the region affected by the T gates is $n_T\,\xi \sim n_T\tau$. We expect that the crossover to GUE statistics should occur when the footprint of the affected region covers all qubits, i.e., when $n_T\tau\sim N$. This relation explains why the larger $n_T$, the shorter time it takes to reach the asymptotic (GUE) value of $\langle \widetilde{r} \rangle$.

(2.3). Where the T gates are inserted matters. Suppose the T gates are inserted at the very beginning of the circuit; then the T gates only change the single qubit phases in the initial random product state, and the resulting entanglement spectrum would be Poisson distributed. We chose to insert the T gates after the entanglement entropy reaches a volume law scaling with the system size to remove any boundary contribution that would complicate the study. That said, a systematic study of the entanglement statistics as function of $n_T$, the depth in which the T gates are inserted, and the system size would be interesting to carry out in future work.

(2.4). To our knowledge this representation of the state has never been studied before. It is a visualization tool of the magnitude of the many-body wavefunction $\Psi(x_A,x_B)$. If one does not take the magnitude and think of the image as a matrix, its singular values give the entanglement spectrum for a bipartition $A/B$. Much as entanglement depends on basis, the details of the image will also depend on basis. For Haar random states the amplitudes are random complex numbers, and therefore the color plot should be structureless in any basis. For the states resulting from evolution by Clifford gates, the apparent structure does not depend on bases related by single-qubit rotations, e.g. the $x$-basis as opposed to the $z$-basis that we choose.

(2.5). We largely agree, but unfortunately we have not found a good handle to study the entanglement level statistics at even a ``semi-analytical'' degree. We hope that the additional studies of the propagation of the influence of the inserted T gates now presented in Figs. 1 and 3 of the revised manuscript may provide further intuition, but we cannot provide much more insight regarding the $n_T N$ scaling other than our systematic numerical studies. That it is difficult to gain further intuition perhaps is a reflection that the result is counter-intuitive, which, in turn, is perhaps what makes it also interesting and unexpected!

Minor points:

(1). We thank the referee for catching this typo. We have fixed this in the revised manuscript.

(2). In the revised manuscript, we have added the circuit depth in the caption of Fig. 1.

(3). We have removed the repetition in the revised manuscript.

[4] T. Rakovszky, S. Gopalakrishnan, S. Parameswaran, and F. Pollmann, “Signatures of information scrambling in the dynamics of the entanglement spectrum,” arXiv preprint arXiv:1901.04444 (2019).

Attachment:

figures_55clWdV.pdf

---

## Round 2 · Referee Report · Anonymous (Referee 3) · 2020-11-3

Report

The authors have significantly clarified the text and I am happy to recommend publication. The paper demonstrates some interesting phenomena that may spur further studies.

---

## Round 2 · Referee Report · Anonymous (Referee 4) · 2020-11-15

Report

The authors have significantly extended and improved their paper. However, I still have some concerns about one aspect of their interpretation of their results, brought up in their response to my previous report (and also discussed in the updated version of the paper). They claim that the time scale for saturating to WD statistics (after the insertion of the T gates) is given by the time needed for the T gates to spread over the entire system. I have various issues with this claim:
- The authors motivate this conjecture by pointing to Ref. 15 of their paper. However, in that reference, the connection to operator spreading arises when considering a subsystem with two edges; the time scale is related to when the edges cease to be independent. For a subsystem consisting of half of an open chain (which I believe is the setup considered in the present paper, although I couldn't find an explicit statement to that effect) the more relevant reference would be Ref. 16, where a saturation to RMT statistics after an O(1) time was observed.
- If the conjecture is correct, then the time needed to achieve WD statistics for n_T = O(1) T-gates is infinite in the thermodynamic limit. Does that mean that there is an issue with the order of limits taken? I.e. that the statement of the paper applies only if the long-time limit is taken first, before the thermodynamic limit? What is the value obtained in the opposite (arguably more physical) limit?

It seems to me that since these issues concern what is arguably the central claim of the paper, they should be clarified before publication. In particular, seeing data for the complementary situation compared to figure 3 (i.e. keeping n_T fixed but varying N) would be helpful in that regard.

  • validity: -
  • significance: -
  • originality: -
  • clarity: -
  • formatting: -
  • grammar: -

Author:  Shiyu Zhou  on 2020-12-02  [id 1068]

(in reply to Report 2 on 2020-11-15)
Category:
answer to question
pointer to related literature

1). We point out that Ref. 16 finds that the level repulsion develops between consecutive non-zero eigenvalues of the reduced density matrix after $\mathcal{O}(1)$ time, when the reduced density matrix still has very low rank. The timescale for the full distribution of the entanglement spectrum to saturate to random matrix theory is set by the operator spreading, consistent with Ref. 15. In our work, past the first battery of Clifford gates but prior to the insertion of T gates, we have a maximally entangled state, with a full-ranked reduced density matrix and a Poisson-distributed entanglement spectrum, and then we look at the level spacing ratios of the full entanglement spectrum following the subsequent time evolution. Therefore, the average $\langle r \rangle$ will only reach that of a Wigner-Dyson distribution when a significant fraction of the entire entanglement spectrum has settled to random matrix theory. Hence, the relevant timescale in our setup should be controlled by the operator spreading, in agreement with Refs. 15 and 16.

2). We concur with the referee that there is indeed an issue with order of limits of time and system size; but the issue is no different from that encountered in phase ordering kinetics. For example, a ferromagnet quenched below its critical temperature will never equilibrate to a uniform magnetization state if the thermodynamic limit is taken first (see Ref. 1 below). The domain sizes of different broken symmetry states will grow with time, but would never reach the system size if the latter is infinite. As long as the domain size grows as a power law in time, one reaches a symmetry broken state in large but finite systems in experimentally observable times (polynomial in the system size). (Once the system settles into a symmetry broken state, reversing the magnetization requires time exponential in the system size). Therefore, as in phase ordering kinetics, to reach the conclusions in our paper one must take the limit of both time and system size to infinity but keeping a power law relation between time and size.

3). ** Figure is in the attached file**
As requested by the referee, we provide the data for $\langle \widetilde{r}(\tau) \rangle$ for the case in which we fix $n_T = 8$ and vary $N = 12, ~14, ~16, ~18$ (see the attached figure). We see that all four curves collapse to a single one when plotted against $\tau \, N^{-1}$, complementing our argument that the action of the inserted T gates is contained within a light-cone, such that each T gate covers a spatial region of size $\xi \sim \tau$ at time $\tau$. At fixed $n_T$, the larger the system size is, the longer time it needs for the effect of $n_T$ T gates to spread over the whole system.

[1] A. J. Bray, “Theory of phase-ordering kinetics,” Advances in Physics 43, 357–459 (1994).

Attachment:

12to18N-t-collapse-nT8.pdf

---

## Round 2 · Author Response

We thank the referees for their helpful comments and suggestions, and have revised our paper accordingly.

---

## Round 2 · List of Changes

We made changes throughout our paper to address referees' comments, and we list the major ones below:

- To avoid any confusion, we removed the differential equation associated to Eq. 2, and the discussion of its flows to the fixed points.
- We modified the scaling exponent $\alpha$ from 0.5 to 0.6 in Eq. 6 for $U'_{\rm Cl} = U^{-1}_{\rm Cl}$ case. An exponent of $\alpha = 0.6$ is consistent with the numerical data, and we realized upon addressing the referee 1's question that this value provides a better fit than the original 0.5.
- We modified discussions related to Fig. 8 in the Conclusions to clarify our proposal of an alternative construction of quantum circuits by concatenating segments of Clifford evolutions with very few T gates inserted at largely spaced layers, upon the request of referee 1.
- We revised last paragraph in the Conclusions to clarify our conjectures upon the request of referee 1.
- We added a new study of the time dependence of $\langle \widetilde{r} \rangle$ past the insertion of the T gates, in a 1D bit array, in answering referee 2's question of how long it takes to reach the GUE limit for $\langle \widetilde{r} \rangle$. We added 1 paragraph at the end of the Introduction, 1 paragraph at Section 3.2, and a new Fig. 3.
- We added a cartoon picture in Fig. 1 (right panel) of how the ES switches from Poisson to Wigner- Dyson distributed with the spreading of the downstream effects of the inserted T gates.

---

## Round 3 · Author Response

We thank Referee 1 for recommending publication of our manuscript. We address comments from Referee 2 in detail in the reply.

---

## Editorial Decision

published